# Decreased Expression of Estrogen Receptors Is Associated with Tumorigenesis in Papillary Thyroid Carcinoma

**DOI:** 10.3390/ijms23031015

**Published:** 2022-01-18

**Authors:** Chen-Kai Chou, Shun-Yu Chi, Yi-Yung Hung, Yi-Chien Yang, Hung-Chun Fu, Jia-He Wang, Chueh-Chen Chen, Hong-Yo Kang

**Affiliations:** 1Division of Endocrinology and Metabolism, Department of Internal Medicine, Kaohsiung Chang Gung Memorial Hospital, College of Medicine, Chang Gung University, Kaohsiung 83301, Taiwan; chou@cgmh.org.tw (C.-K.C.); kid1412310@gmail.com (J.-H.W.); masaki1225151@yahoo.com.tw (C.-C.C.); 2Department of Surgery, Kaohsiung Chang Gung Memorial Hospital, College of Medicine, Chang Gung University, Kaohsiung 83301, Taiwan; abraban@cgmh.org.tw; 3Department of Psychiatry, Kaohsiung Chang Gung Memorial Hospital, College of Medicine, Chang Gung University, Kaohsiung 83301, Taiwan; ian670523@cgmh.org.tw; 4Department of Dermatology, Kaohsiung Chang Gung Memorial Hospital, College of Medicine, Chang Gung University, Kaohsiung 83301, Taiwan; yichienyang@gmail.com; 5Department of Obstetrics and Gynecology, Kaohsiung Chang Gung Memorial Hospital, College of Medicine, Chang Gung University, Kaohsiung 83301, Taiwan; allen133@cgmh.org.tw; 6Graduate Institute of Clinical Medical Sciences, Chang Gung University, Kaohsiung 83301, Taiwan; 7Center for Hormone and Reproductive Medicine Research, Department of Obstetrics and Gynecology, Kaohsiung Chang Gung Memorial Hospital, College of Medicine, Chang Gung University, Kaohsiung 83301, Taiwan

**Keywords:** papillary thyroid carcinoma, estrogen receptors, tumorigenesis, sex difference

## Abstract

Papillary thyroid carcinomas (PTC), which is derived from thyroid follicular cells, is the most commonly differentiated thyroid cancer with sex disparity. However, the role of estrogen receptors (ERs) in the pathogenesis of PTC remains unclear. The present study aimed to determine the association of ER mRNA expression levels with clinicopathologic features in PTC. To that aim, the mRNA levels of *ESR1* (ERα66), *ESR1* (ERα36), *ESR2*, and G-protein-coupled estrogen receptor 1 (*GPER1*) in snap-frozen tissue samples from PTCs and adjacent normal thyroid tissues were determined using quantitative reverse transcription polymerase chain reaction (RT-qPCR), and the correlation between ER mRNA expression levels and clinicopathologic features was analyzed. The expression of ERα66, ERα36, ERβ, and *GPER1* was lower in PTC specimens than in adjacent normal thyroid tissues. Moreover, low *GPER1* expression was associated with extrathyroidal extension. There was no obvious difference in expression of ERs between PTC specimens from male and female patients. In conclusion, our findings highlight the importance of ERs in PTC tumorigenesis.

## 1. Introduction

Papillary thyroid carcinoma (PTC), which is derived from thyroid follicular cells, is the most common form of well-differentiated thyroid cancer [1]. The prevalence of PTC has increased in recent decades, with female cases accounting for as many as 60–80% of the PTC cases in different populations and geographical areas [2]. Whilst women have an increased incidence of thyroid tumors, male sex is an unfavorable prognostic factor in PTC [3,4]. Therefore, sex hormones and the consequences derived from their associated signaling may play pivotal roles in the pathogenesis of PTC. Many advances in the diagnosis and management of PTC have been made; however, the influence of molecular factors on the development of PTC in different genders and the association of these factors with clinicopathological features remain to be elucidated.

Sex hormone receptors are a group of steroid hormone receptors including androgens, estrogen, and progesterone [5]. Our previous study [6] investigated the impact of androgen receptor (AR) gene expression on the clinical features and progression of PTC. Our findings suggested that sex steroids and their receptors may play an important role in PTC tumorigenesis. Classical estrogen signaling is mediated through estrogen receptors (ERs), which are members of a large family of nuclear transcription factors [7]. The estrogen receptors α (ERα) and β (ERβ) are two distinct ERs encoded by *ESR1* (NR3A1) and *ESR2* (NR3A2), respectively [8,9]. E2 and the estrogen receptors, ERα and ERβ, have also been shown to participate in the pathology of thyroid cancer [10,11]. ERβ was demonstrated as having a protective role in carcinogenesis by regulating antiproliferative and proapoptotic signals, and loss of ERβ leads to unfavorable prognosis in thyroid cancers [12,13,14,15]. However, their expression patterns vary considerably, which hinders the identification of their role in the pathogenesis of thyroid cancer. Furthermore, recent studies have focused on ERα36, a novel 36 kDa variant of the traditional full-length ERα, and the transmembrane estrogen receptor G-protein-coupled estrogen receptor 1 (*GPER1*). Nevertheless, although the importance of ERs in breast and prostate cancers is well established [16,17], little is known about the role of these ERs in tumorigenesis of PTC.

A better understanding of the clinical relevance of ERs in PTC initiation and progression may aid in the identification of novel therapeutic targets and molecular markers. However, there is limited and varying information on the relationship between ER expression and its correlation with clinical features in male and female PTC cases. Here, we simultaneously examined *ESR1* (ERα66), *ESR1* (ERα36), *ESR2*, and *GPER1* mRNA expression levels in PTC specimens using quantitative reverse transcription polymerase chain reaction (RT-qPCR), and we assessed the correlation between ER expression and clinicopathologic characteristics in PTC.

## 2. Results

### 2.1. Patient Characteristics

This prospective study included 103 patients with PTC (82 women and 21 men) who received standard treatment, including surgery, radioactive iodine therapy, and thyroid hormone therapy, between August 2019 and July 2021 at the Kaohsiung Chang Gung Memorial Hospital, Kaohsiung, Taiwan. The clinicopathological features of these cases are shown in Table 1.

### 2.2. ER Expression Is Lower in PTC Specimes Than in Adjacent Normal Thyroid Tissues

The expression of *ESR1* (ERα66), *ESR1* (ERα36), *ESR2*, and *GPER1* was analyzed in these samples by RT-qPCR. As shown in Figure 1, the levels of all mRNAs were significantly lower in PTC specimens than in the surrounding normal parenchyma (Figure 1A–D). A cycle threshold (Ct) value > 40 was defined as “not detectable” in this study; undetectable Ct values were only observed in PTC tumor specimens. Notably, only 16.5%, 19.4%, 30.1%, and 33% of the PTC samples showed upregulated expression of ERα66, ERα36, ERβ, and *GPER1*, respectively. In contrast, the expression of ERα66, ERα36, ERβ, and *GPER1* was classified as downregulated or undetectable in 83.5%, 80.6%, 69.9%, and 67% of all samples (Figure 2A–D).

We further analyzed ER expression according to different gender categories. The mRNA expression levels of ERα66 and ERα36 were significantly (*p* < 0.05) lower in PTC tumor specimens than in adjacent normal tissues in both female and male patients (Figure 3). Moreover, the mRNA expression levels of *ESR2* and *GPER1* were also significantly lower in PTC tumor specimens than in adjacent normal tissues in female patients. However, there was no obvious difference in ER expression between PTC specimens when comparing female and male patients (Figure 3).

We then analyzed the GDC TCGA Thyroid Cancer (THCA) dataset describing ER expression in PTC specimens, which is publicly available through the University of California Santa Cruz (UCSC) Xena Browser [18]. These data further confirmed that the expression levels of ERα, ERβ, and *GPER1* are lower in PTC specimens than in normal human thyroid tissues (Appendix A). We also evaluated data from microarray-based RNA profiling analysis using the Gene Expression Omnibus (GEO) datasets GSE6004, GSE165724, and GSE153659; the findings further confirmed that ER expression is lower in PTC specimens than in normal human thyroid tissues (Appendix A). In agreement with our results (Figure 3), we observed significantly lower mRNA expression levels of *ESR2* and *GPER1* in PTC specimens from both male and female patients in the GSE165724 and GSE153659 datasets (Appendix A). However, data on the expression of specific ERα variants (ERα66 and ERα36) are not available from these datasets without accession numbers.

### 2.3. Correlation of ER Expression with Clinicopathological Characteristics in PTC

The correlations between *ESR1* (ERα66), *ESR1* (ERα36), *ESR2*, and *GPER1* mRNA expression levels and the demographic and clinical characteristics of the patients were further assessed. *GPER1* mRNA expression level was significantly lower in patients with extrathyroidal extension than in those without (*p* < 0.05; Table 2). However, the patterns of ER mRNA expression level did not differ significantly among different sex and age groups, tumor stages, or patients with and without lymph node metastasis (Table 2).

To investigate whether ER expression is involved in thyroid cancer cell tumorology, the mRNA expression levels of ERα66, ERα36, ERβ, and *GPER1* were assessed in various thyroid cancer cell lines and a normal thyroid cell line. As demonstrated in Figure 4, their expression was significantly lower in PTC cancer cell lines (TPC-1 and MDAT) and anaplastic thyroid cancer (ATC) cell line (8505C) than in the normal thyroid cell line (Nthy-ori-3-1). This result suggests that decreased ER expression may play an important role in thyroid tumorigenesis and dedifferentiation.

## 3. Discussion

Estrogens are steroid hormones that play a key role not only in the regulation of reproductive organs, but also in tumor biology, including cell growth and differentiation in both males and females [9]. In the present study, we determined the mRNA expression levels of ERs in PTC specimens and examined the correlation with clinicopathological characteristics. In agreement with our previous study [6] which enrolled 71 PTC patients, we found that not only the mRNA expression levels of ERα66 and ERβ but also the expression levels of ERα36 and *GPER1* were all decreased in PTC specimens compared to that in paired adjacent normal tissues. Furthermore, low *GPER1* mRNA expression level was associated with extrathyroidal extension. These results suggest that, in addition to a decrease in ERβ expression [19], the decreased expression of other ERs is a common step in PTC tumor development.

Previous immunohistochemical studies have shown that ERα expression is greater in PTC tumors than in normal thyroid tissues, whereas ERβ expression is significantly lower in neoplastic than in nonneoplastic thyroid tissues [20,21]. Decreased ERβ mRNA and protein expression has been associated with the occurrence of nodular hyperplasia and PTC [22]. In contrast, there are contradictory reports regarding ERα and ERβ expression patterns, which have shown wide variations and discrepancies among studies as a result of different experimental methods, study designs, and sample sources [9]. The investigation of tumor biology, irrespective of the methodology employed, requires appropriate study controls. Using normal adjacent tissue as a control, as in this study, has many advantages over paired tumor tissue, such as the relative minimization of the variability and selection bias between study individuals. Previous studies have shown that DNA methylation may silence the promoter of *ESR1* in several cancers [23,24]. For example, DNA hypermethylation is a major reason for the loss of ERα expression and confers a poor prognosis for breast cancer [25]. It is possible that epigenetic modifications of *ESR1*, such as DNA hypermethylation, may play an important role in PTC tumorigenesis. However, this hypothesis remains to be investigated in PTC.

A novel nonclassical ER pathway involves *GPER1*, formerly known as GPR30, which binds estrogen with high affinity and functions alongside the traditional nuclear ERs to regulate cellular and physiological responsiveness to estrogen [26]. In addition, ERα36 is believed to be involved in tumor behavior; it lacks both transcriptional activation domains and is located in the cytoplasm, where it mediates nongenomic estrogen signaling. Several studies have discussed the role of these novel ERs in the regulation of thyroid follicular cells in thyroid disease. A clinical study demonstrated that *GPER1* expression is lower in goiter specimens than in normal thyroid tissues, suggesting that *GPER1* may be involved in the pathogenesis of goiter [27]. ERα36 can crosstalk with the MAPK/ERK pathway and promote cell aggressiveness [28]. Here, we demonstrated that the mRNA expression level of *GPER1* and ERα36 is decreased in PTC specimens compared to that in paired adjacent normal tissues. Our findings are in agreement with a recent study reporting the tumorigenesis role of decreased *GPER1* expression according to in silico analyses of the GEO and TCGA databases [29]. Furthermore, in agreement with our findings showing that low *GPER1* expression is associated with extrathyroidal extension, there is evidence that the activation of *GPER1* inhibits epithelial–mesenchymal transition (EMT), which is accompanied by upregulation of E-cadherin and downregulation of N-cadherin and vimentin in goat mammary epithelial cells [30].

Recent studies have addressed the influence of ERs on thyroid cancer; however, the results have been inconsistent. Previous studies have demonstrated that higher ERα expression enhances cell proliferation. Furthermore, ER expression has been shown to be correlated with tumor size and the proliferation marker Ki-67 in PTC, reaffirming the role of ERα in tumor growth [31]. However, a significantly higher proportion of tumors from disease-free patients have been shown to be ERα positive, indicating an inverse relationship between ERα expression and poor clinical outcomes [32]. In the current study, the pattern of ER mRNA expression levels did not differ significantly among different age groups, tumor stages, or patients with and without lymph node metastasis (Table 2). Similar to a previous study [33], we also found no significant differences in the expression levels of ERs in PTC samples from men and women with PTC. These data suggest that the expression level of ERs may not be directly involved in the disparities in PTC prevalence and progression between men and women. It is possible that the current cross-sectional study is limited by sample size and follow-up interval. The lack of longitudinal and prognostic factors hinders the determination of the role of ERs in PTC. These limitations could be overcome with larger sample sizes and adequate follow-up periods.

## 4. Materials and Methods

### 4.1. Tumor Samples and Patient Information

This is a prospective study composed of PTC patients treated between August 2019 and July 2021 at the Kaohsiung Chang Gung Memorial Hospital, Kaohsiung, Taiwan. We omitted those with follicular carcinoma, medullary cancer or anaplastic cancer. Patients aged less than 18 years or more than 80 years and those without an adequate follow-up period were also excluded. Tissue samples were snap-frozen in liquid nitrogen at the time of total thyroidectomy and subsequently stored in liquid nitrogen. Details concerning clinical data collection and tumor node metastasis classification for these samples were previously described elsewhere [34]. Patients that were aged <55 years with stage I PTC and those that were aged <55 years with stage I or II PTC were defined as the low-risk group according to the American Joint Commission on Cancer-International Union Against Cancer criteria [35]. The remaining patients were defined as the high-risk group.

### 4.2. RNA Extraction and Quantitative Reverse Transcription Polymerase Chain Reaction (RT-qPCR)

Total RNA was extracted from surgical specimens (≤50 mg) stored in DNA/RNA ShiledTM (ZYMO, Irvine, CA, USA) and thyroid cancer cell lines using Quick-RNATM Mini Prep kit (ZYMO, Irvine, CA, USA). cDNA templates were obtained by reverse-transcribing 2 µg of total RNA using the M-MLV Reverse Transcriptase Kit (Promega, Madison, WI, USA). The reaction conditions for the procedure were as follows: 70 °C for 10 min with 2 µg of RNA, 10 µM of Oligo dT and ddH_2_O; 42 °C for 1 h; 95 °C for 1 min with the previous mix and 5× MMLV RT buffer, 2.5 mM dNTP, MMLV Reverse Transcriptase, and RNasin. The primers for real-time PCR were as follows: 18S rRNA, forward (5′–GTAACCCGTTGAACCCCATT–3′) and reverse (5′–CCATCCAATCGGTAGTAGTG–3′); ERα36, forward (5′–CCAAGAATGTTCAACCACAACCT–3′) and reverse (5′–GCACGGTTCATTAACATCTTTCTG–3′); ERα66, forward (5′–AAGAAAGAACAACATCAGCAGTAAAGTC–3′) and reverse (5′–GGGCTATGGCTTGGTTAAACAT–3′); ERβ, forward (5′–TACTGACCAACCTGGCAGACAG–3′), forward (5′–AGCCGGTCCGGGTGCAAG–3′) and reverse (5′–CCACCCAGAGCCCGAGGG–3′); GPER1, forward (5′–AGTCGGATGTGAGGTTCA–3′), and reverse (5′–TCTGTGTGAGGAGTGCAA–3′). The RT-qPCR was performed with 2 µL of ten-fold-diluted cDNA using 7 µL of Fast SYBR Green Master Mix (Applied Biosystems, Life Technologies, New York, NY, USA) and 0.5 µL of 10 uM forward and reverse primers. The reaction conditions for the procedure were as follows: 95 °C for 20 s, followed by 40 cycles of 95 °C for 3 s and 60 °C for 30 s. The protocol for RT-qPCR was according to the manufacturer’s instructions.

### 4.3. Thyroid Cancer Cell Culture

Three human thyroid cancer cell lines (TPC1, MDA-T32, and 8505C) and one normal thyroid cell line (Nthy-ori-3-1) were used in this study. TPC1, MDA-T32 (American Type Culture Collection, cat#30-2001, Manassas, VA, USA), and normal cell line Nthy-ori-3-1 (European Collection of Authenticated Cell Cultures (ECACC), Porton Down, UK) were routinely cultured in RPMI1640 containing 10% fetal bovine serum (FBS), 100 units/mL penicillin, and 2 mM l-glutamine (Gibco, Carlsbad, CA, USA) at 37 °C in a humidified chamber containing 5% CO_2_. The 8505C cells (ECACC, Porton Down, Porton, UK) were routinely cultured in MEM containing 10% fetal bovine serum (FBS), 100 units/mL penicillin, and 1% sodium pyruvate.

### 4.4. Bioinformatics Analysis

The bioinformatics data on the ERs expression of PTC were publicly available from UCSC Xena Browser under GDC THCA datasets. The chosen phenotypic parameters were the sample type, gender, and histological type, which allowed easily selecting the PTC classical normal and tumor regions, male and female datasets, and genes of interest (*ESR1*, *ESR2*, and *GPER1*), before downloading the files for analysis. We also analyzed gene expression data from three microarray datasets (GSE6004, GSE165724, and GSE153659) from GEO datasets. The raw data file of this array was downloaded, and the values of the ERs were analyzed.

### 4.5. Statistical Analyses

Continuous variables, including ER expression in paired cancer and normal tissue from patients with PTC, were compared using Wilcoxon test. Correlations between ER expression levels and tumor characteristics were assessed by Mann–Whiney U-test analysis. All statistical analyses were performed using SPSS software (version 19.0; SPSS, Chicago, IL, USA). All *p*-values were two sided, with *p*-values < 0.05 considered statistically significant.

## 5. Conclusions

Our findings demonstrate that PTC specimens have reduced ER expression and that *GPER1* expression is associated with extrathyroidal extension. These findings highlight the importance of ERs in the tumorigenesis and clinical presentation of PTC. Future studies with large sample sizes and appropriate follow-up periods will help to elucidate the association of ERs expression with overall survival and PTC-related death. This may facilitate personalized patient care by using the expression profiling of ERs as a prognostic factor to guide follow-up and management decisions.

## Figures and Tables

**Figure 1 ijms-23-01015-f001:**
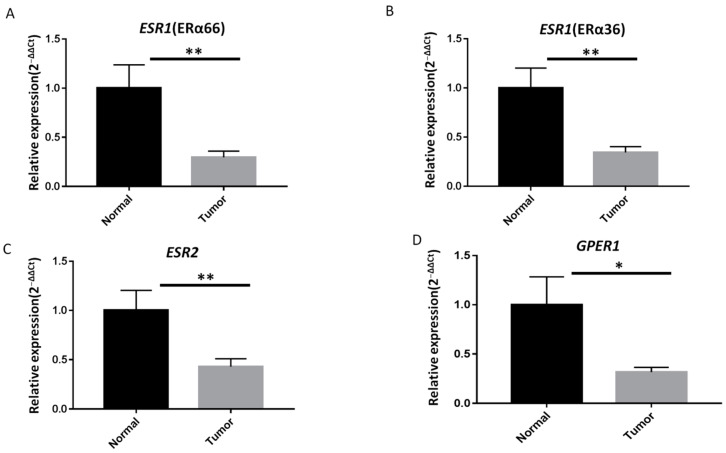
ER mRNA expression is lower in papillary thyroid carcinoma (PTC) specimens than in adjacent normal thyroid tissues. Quantitative reverse transcription polymerase chain reaction (RT-qPCR) analysis of (**A**) full-length estrogen receptor 1 (ERα66), (**B**) 36 kDa version of *ESR1* (ERα36), (**C**) ERβ, and (**D**) G-protein-coupled estrogen receptor 1 (*GPER1*) mRNA expression levels in PTC specimens (*n* = 103) and paired adjacent normal thyroid tissues. The fold change values indicate the relative change in the expression levels between samples and the internal control (18S rRNA), assuming that the expression level of 18S rRNA in each sample was set to 1. Relative mRNA expression levels in PTC specimens were normalized to adjacent normal tissues (T/N fold change). The mRNA expression levels of ERα66, ERα36, ERβ, and *GPER1* were significantly lower in tumor tissues than in matched adjacent normal tissues. Fold change in mRNA expression levels in tumor samples was calculated relative to that in paired normal thyroid tissue; ** *p* < 0.005, * *p* < 0.05.

**Figure 2 ijms-23-01015-f002:**
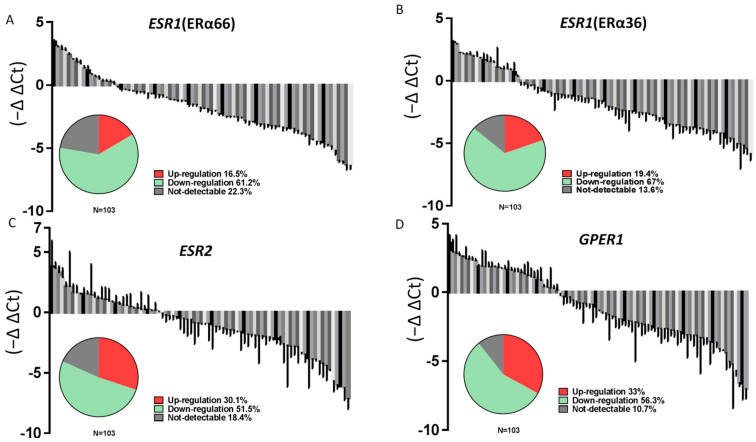
The majority of patients with PTC show lower ERs expression in cancer specimens than in adjacent normal thyroid tissues. Relative mRNA expression levels of (**A**) ERα66, (**B**) ERα36, (**C**) ERβ, and (**D**) *GPER1* in PTC (*n* = 103) samples normalized to that in adjacent normal tissues (T/N fold change). The pie charts show the proportion of samples with upregulated, downregulated, and undetectable expression.

**Figure 3 ijms-23-01015-f003:**
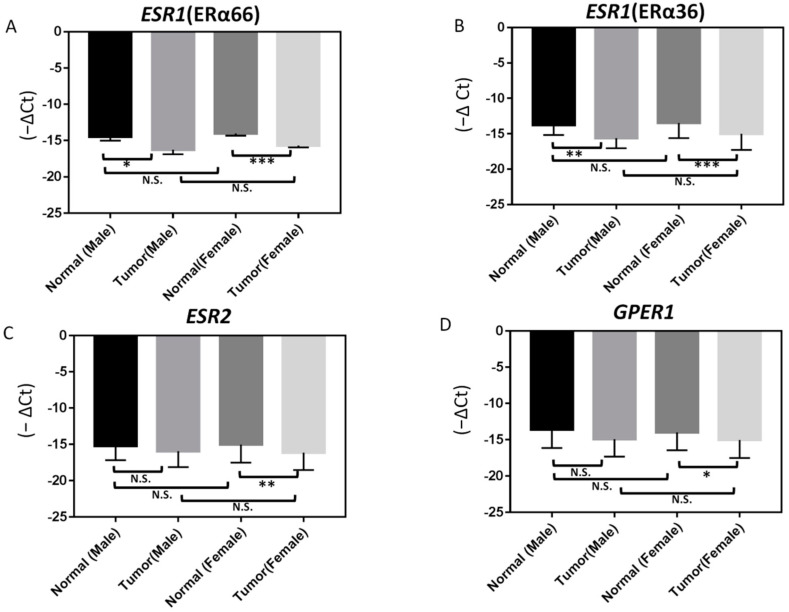
Differences in ERs expression between normal and papillary thyroid carcinoma (PTC; tumor) samples from male and female patients as determined by RT-qPCR: (**A**) ERα66, (**B**) ERα36, (**C**) ERβ, and (**D**) *GPER1*. In samples from female patients, the expression of ERβ and *GPER1* is significantly lower in PTC specimens than in adjacent normal tissues. In contrast, the expression of ERα66 and ERα36 in PTC specimens is significantly lower than in adjacent normal tissues in both male and female patients; *** *p* < 0.001, ** *p* < 0.005, * *p* < 0.05. N.S.: not significant.

**Figure 4 ijms-23-01015-f004:**
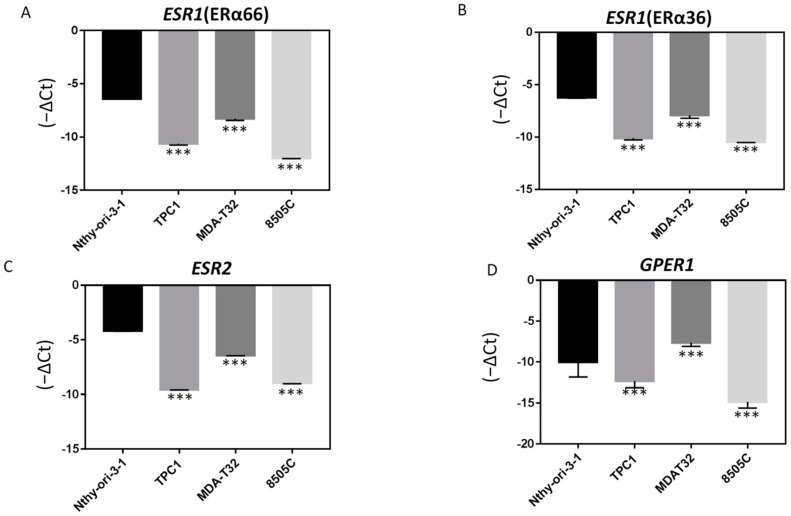
Endogenous ER mRNA expression levels are higher in PTC-derived cancer cells than in normal thyroid cells. The mRNA expression level of (**A**) ERα66, (**B**) ERα36, (**C**) ERβ, and (**D**) *GPER1* in the normal thyroid cell line, Nthy-ori-3-1, is higher than that in PTC-derived cancer cell lines (TPC1 and MDAT32) and the ATC-derived cancer cell line (8505C); *** *p* < 0.001 compared with normal thyroid cells.

**Table 1 ijms-23-01015-t001:** Clinicopathological features of the papillary thyroid carcinoma (PTC) cases examined in this study (*n* = 103).

Clinical Features	*n* (%)
Age at the time of diagnosis (years)	47.18 ± 12.82
Sex (male/female)	21/82
Tumor size (cm)	1.90 ± 1.10
Lymph node metastasis	38 (36.8%)
Extrathyroidal extension	30 (29.1%)
Tumor staging (AJCC) ^a^	
Low-risk	93 (90.2%)
High-risk	10 (9.7%)
Distant metastasis	1 (0.97%)

^a^ The low-risk group consisted of patients aged ≤55 years with stage I PTC and those aged >55 years with stage I or II PTC according to the American Joint Committee on Cancer (AJCC). The remaining patients were defined as the high-risk group.

**Table 2 ijms-23-01015-t002:** Association of estrogen receptor (ER) mRNA expression levels with the clinicopathological features of papillary thyroid carcinoma (PTC) cases (*n* = 103).

Clinical Features		ERα66		ERα36		ERβ		*GPER1*	
		2^−(ΔΔCt)^	*p*	2^−(ΔΔCt)^	*p*	2^−(ΔΔCt)^	*p*	2^−(ΔΔCt)^	*p*
Age (years)	<55	1.77 ± 0.28	0.60	1.94 ± 0.22	0.59	1.11 ± 0.29	0.09	1.89 ± 0.31	0.63
	≥55	1.23 ± 0.21		2.49 ± 0.44		0.66 ± 0.14		2.31 0.83	
Sex	Feale	1.67 ± 0.23	0.47	1.44 ± 0.23	0.06	1.10 ± 0.24	0.44	1.68 ± 0.25	0.40
	Male	1.25 ± 0.30		0.8 ± 0.43		0.99 ± 0.22		2.24 ± 0.64	
Lymph node metastasis	Absent	1.43 ± 0.24	0.36	1.95 ± 0.20	0.49	1.21 ± 0.35	0.74	2.22 ± 0.50	0.70
	Present	1.84 ± 0.33		1.76 ± 0.42		0.95 ± 0.15		1.79 ± 0.39	
Extrathyroidal extension	Absent	1.76 ± 0.27	0.79	1.06 ± 0.22	0.16	0.83 ± 0.12	0.12	2.2 ± 0.45	0.04 ^b^
	Present	1.03 ± 0.16		1.8 ± 0.45		1.65 ± 0.62		0.97 ± 0.22	
Tumor staging (AJCC) ^a^	Low	1.62 ± 0.22	0.60	1.1 ± 0.20	0.11	1.09 ± 0.23	0.76	1.86 ± 0.33	0.19
	High	0.95 ± 0.23		2.7 ± 0.91		1.05 ± 0.36		0.90 ± 0.73	

^a^ The low-risk group consisted of patients aged ≤55 years with stage I PTC and those aged >55 years with stage I or II PTC according to the American Joint Committee on Cancer (AJCC). The remaining patients were defined as the high-risk group. ^b^
*p* < 0.05.

## Data Availability

The raw data supporting the conclusions of this article will be made available by the authors, without undue reservation.

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
