# Peer review of "Decreased Expression of Estrogen Receptors Is Associated with Tumorigenesis in Papillary Thyroid Carcinoma"

_ijms, 2022, doi:10.3390/ijms23031015_

Round 1
Reviewer 1 Report
In the study titled ‘Decreased Expression of Estrogen Receptors is associated with 2
Tumorigenesis in Papillary thyroid Carcinoma by Chou et.al., have investigated the expression of Estrogen Receptors in the pathogenesis of papillary thyroid carcinomas. Authors have studied and compared the expression profile of ERα66, ERα36, ERβ and GPER1 in patient derived tumor tissues and adjacent normal thyroid tissues by RT-qPCR method. The major finding of this study is that the expression of ERs and GPER1 is lower in PTC samples than in adjacent normal tissues. Authors have also reported that low GPER1 mRNA expression is associated with extrathyroidal extension. Overall the manuscript is interesting and well written and may be considered for publication in IJMS.
Author Response
Tumorigenesis in Papillary thyroid Carcinoma by Chou et.al., have investigated the expression of Estrogen Receptors in the pathogenesis of papillary thyroid carcinomas. Authors have studied and compared the expression profile of ERα66, ERα36, ERβ and GPER1 in patient derived tumor tissues and adjacent normal thyroid tissues by RT-qPCR method. The major finding of this study is that the expression of ERs and GPER1 is lower in PTC samples than in adjacent normal tissues. Authors have also reported that low GPER1 mRNA expression is associated with extrathyroidal extension. Overall the manuscript is interesting and well written and may be considered for publication in IJMS.
Reply: Thanks for your comment.
Reviewer 2 Report
Chou et al. analyse expression levels of genes encoding estrogen receptors (ERs) in papillary thyroid cancer (PTC) using RT-qPCR. I recommended rejection of this article because of the low novelty. My major remarks are listed below.
- The introduction does not fully describe the current state of knowledge about ERs expression in PTC.
- The Authors state in lines 170-172 that „To the best of our knowledge, this is the first report on the role of ERs in the tumorigenesis and clinical presentation of PTC”. However, ERα66 and ERβ levels were already analysed in PTC and peri-tumoral thyroid tissues using RT-qPCR in the Authors’ former paper published in Cancers (Chou et al., Cancers 2020, 12(5), 1109). Nevertheless, it is not mentioned in the current manuscript.
- Materials and Methods (Section 4.1. and 4.2.): It should be clarified if tissues were paraffin-embedded before RNA isolation, as mentioned in the Abstract. In Materials and Methods it is only stated that tissues were snap frozen, stored in liquid nitrogen and then RNA was isolated.
- Materials and Methods (Section 4.2.): The Authors should provide information on (i) the weight of the tissue used to isolate RNA, (ii) cDNA reverse transcription (name of the kit used, supplier, amount of RNA per reaction, etc.), (iii) RT-qPCR protocol (e.g., primers concentration, dilution factor for cDNA, amplification program details).
Author Response
1.The introduction does not fully describe the current state of knowledge about ERs expression in PTC.
Reply: Thanks for your comment. The introduction about ERs expression in PTC has been re-written in line 57-62. The revised content has been more informative that we feel have strengthened the manuscript as reviewer suggested.
2.The Authors state in lines 170-172 that „To the best of our knowledge, this is the first report on the role of ERs in the tumorigenesis and clinical presentation of PTC”. However, ERα66 and ERβ levels were already analysed in PTC and peri-tumoral thyroid tissues using RT-qPCR in the Authors’ former paper published in Cancers (Chou et al., Cancers 2020, 12(5), 1109). Nevertheless, it is not mentioned in the current manuscript.
Reply: Thanks for your comment. We agree with reviewer’s point and have deleted the above statement in line 177-178 and we also mentioned our previous study that enrolled 71 PTC patients. In this study, we has increased the subject numbers and analyzed the ERs (ESR1(ERα66), ESR1(ERα36), ESR2 and GPER) expression levels as a whole in PTCs to correlate with PTC’s tumor characteristics in line 171-174.
3.Materials and Methods (Section 4.1. and 4.2.): It should be clarified if tissues were paraffin-embedded before RNA isolation, as mentioned in the Abstract. In Materials and Methods it is only stated that tissues were snap frozen, stored in liquid nitrogen and then RNA was isolated.
Materials and Methods (Section 4.2.): The Authors should provide information on (i) the weight of the tissue used to isolate RNA, (ii) cDNA reverse transcription (name of the kit used, supplier, amount of RNA per reaction, etc.), (iii) RT-qPCR protocol (e.g., primers concentration, dilution factor for cDNA, amplification program details).
Reply: Thanks for pointing out our errors. We had corrected this mistake and revised the sentences in Abstract section: “To that aim, the mRNA levels of ESR1(ERα66), ESR1(ERα36), ESR2 and G protein-coupled estrogen receptor 1 (GPER1) in snap frozen tissue samples from PTCs and adjacent normal thyroid tissues” in line 27-29.
Section 4.2.
(i) The weight of tissue used to isolate RNA is around 50mg.
(ii) cDNA reverse transcription cDNA templates were obtained by reverse transcribing 2ug of total RNA using the M-MLV Reverse Transcriptase Kit (Promega). The reaction condition for the procedure were as follows: 70℃ 10mins with the 2ug RNA, 10uM Oligo dT and ddH2O; then 42℃ 1hr, 95℃ 1 mins with previous mix and 5X MMLV RT buffer, 2.5mM dNTP, MMLV Reverse Transcriptase and RNasin.
(iii) The quantitative reverse transcription–polymerase chain reaction (RT-PCR) was performed with 2ul of 10-fold-diluted cDNA using 7ul of Fast SYBR Green Master Mix (Applied Biosystems) and 0.5ul of forward and reverse 10uM primer. The reaction condition for the procedure were as follows: 95℃ 20sec, followed by 40cycles of 95℃ for 3sec and 60℃ for 30sec. We had added the above information in Materials and Methods (Section 4.2.)
Reviewer 3 Report
The authors have written a paper with the title “Decreased expression of ERs is associated with tumorigenesis in papillary thyroid carcinoma”. They have retrospectively analyzed the mRNA levels of ERα66, ERα36, ERβ and GPER1 by quantitative PCR in 103 thyroid cancer patients. They showed that the levels of these receptors were lower in tumors than in adjacent normal thyroid tissues. They have found no association between expression levels and clinicopathological features other than a significant association between low GPER1 expression and ETE.
The samples size, as the authors also state, is one limitation of this paper.
The paper needs extensive English editing. There are several mistakes, as in lines 41, 86, 88, 123, 142, etc…
In the results section, when describing results represented in figure 2 (lines 88-92), the authors should keep the same order when describing the genes otherwise is quite confusing.
The “conclusion” on line 92 is a big overstatement.
Phrase in line 142 does not make sense…. Whether what??
In line 145, isn’t the word anaplastic misplaced??
Table 2 should be improved for better/easier understanding.
Line 165 would be better mRNA expression LEVELS (as in other places where the same is written).
Line 169 doesn’t make sense as it is.
Discussion should be improved. The authors suggest that hypermethylation of ESR1 gene may play an important role in its expression. Why don’t the authors test that? It is a simple experiment to perform.
Round 2
Reviewer 2 Report
The manuscript has been improved, but I still think the study is too preliminary.
I would suggest making the changes listed below.
- The Authors should pay more attention to the correct spelling of gene names. They should always be written in italics (e.g., in lines 33, 118, 129, 333, and all figures).
- ‘18S’ (line 121) and ‘h18S’ (line 283) should be replaced with ‘18S rRNA’.
Author Response
1.The Authors should pay more attention to the correct spelling of gene names. They should always be written in italics (e.g., in lines 33, 118, 129, 333, and all figures).
Reply:Thanks for your comment. We had modified these genes names in italics through the whole manuscript as reviewer’s suggestion in lines 29, 32, 33, 65, 95, 96, 113, 118, 123, 124, 130, 148, 156, 172, 193, 258, 280, 300, 301, 304, 306 and 333.
2.‘18S’ (line 121) and ‘h18S’ (line 283) should be replaced with ‘18S rRNA’.
Reply:Thanks for your comment. We had modified these sentences as reviewer’s suggestion. We had replaced ‘18S’ (line 110, 111) and ‘h18S’ (line 251) with ‘18S rRNA’.
Reviewer 3 Report
Thank you for your reply. The suggestions I have made have been revised. However, there are still some things that should be corrected. I attach a file with some comments included (as note or marked as light yellow).

Author Response
However, there are still some things that should be corrected. I attach a file with some comments included (as note or marked as light yellow).
Reply: Thanks for your comment. We had modified these sentences according to reviewer’s suggestion. All changes to the main text have been marked up in yellow in the revised version.
In line 96-97: We defined a cycle threshold (Ct) value >40 was defined as “not detectable” and undetectable Ct values were only observed in PTC tumor specimens. “not detectable” value was classified as downregulated in the current study.
So the expression of ERα66, ERα36, ERβ, and GPER1 was classified as downregulated or undetectable in 83.5% (61.2+ 22.3 %), 80.6 (67+ 13.6)%, 69.9 (51.5+18.4)%, and 67 (56.3+10.7)% of all samples in line 96-97.